# Associations between Brain Microstructure and Phonological Processing Ability in Preschool Children

**DOI:** 10.3390/children9060782

**Published:** 2022-05-26

**Authors:** Ying Zhou, Guangfei Li, Zeyu Song, Zhao Zhang, Huishi Huang, Hanjun Li, Xiaoying Tang

**Affiliations:** 1School of Life Science, Beijing Institute of Technology, 5 South Zhongguancun Street, Beijing 100081, China; a_zhouying@163.com (Y.Z.); 18910328635@163.com (G.L.); 3120205893@bit.edu.cn (Z.S.); zhangzhao089@sina.cn (Z.Z.); xiaoying@bit.edu.cn (X.T.); 2Department of Medical Physics and Biomedical Engineering, University College London, Gower Street, London WC1E 6BT, UK; huishi.huang.21@ucl.ac.uk

**Keywords:** DTI, white matter, language, development

## Abstract

Neuroimaging studies have associated brain changes in children with future reading and language skills, but few studies have investigated the association between language skills and white matter structure in preschool-aged children. Using 208 data sets acquired in 73 healthy children aged 2–7 years, we investigated the relationship between developmental brain microstructure and phonological processing ability as measured using their phonological processing raw score (PPRS). The correlation analysis showed that across the whole age group, with increasing age, PPRS increased, fractional anisotropy (FA) of the internal capsule and inferior fronto-occipital fasciculus and some other regions increased, and mean diffusivity (MD) of the corpus callosum and internal capsule and some other regions decreased. The results of the mediation analysis suggest that increased FA may be the basis of phonological processing ability development during this period, and the increased number of fiber connections between the right inferior parietal lobule and right supramarginal gyrus may be a key imaging feature of phonological processing ability development. Our study reflects the changes in brain microstructure and contributes to understanding the underlying neural mechanisms of language development in preschool children.

## 1. Introduction

In the first years of life, the microstructure of the brain undergoes a period of rapid maturation. This is followed by a slower rate of brain development into adulthood. One of the most important skills young children acquire as they mature is language. Language skills have been shown to correlate with future reading skills and may influence future mental health, academic achievement, and career prospects [1,2]. The acquisition of language and reading skills occurs dynamically throughout childhood, with the development of brain areas needed for language function beginning in the womb [3]. In the first year of life, babies already learn language through skills such as speech recognition, speech awareness, acoustics, and word segmentation [4]. Pre-reading skills and primary competencies such as speaking and language comprehension improve rapidly and steadily through early childhood (ages 2 to 7) [5,6] and serve as a foundation for reading development [7]. Although most children experience typical neurodevelopment, approximately 7% of the population receive a diagnosis of dyslexia [8]. Improving understanding of typical language development may facilitate the early detection of deviations from this. Language skills reflect a person’s ability to control the sound structure of spoken words. Assessment of language skills, such as the phonological processing scaled score and speed-naming scores, provide effective indicators for identifying children at risk of dyslexia [9,10].

Neuroimaging studies have shown that early brain development is dependent upon a number of complexly interwoven mechanisms, including the maturation and functional specialization of gray matter (GM) regions and the establishment of connections and myelination of white matter (WM) between different brain regions [11]. This anatomical development is accompanied by psychological, motor, cognitive, and linguistic growth, but the relationship between brain maturation and behavioral changes observed during development is not well-understood. Diffusion tensor imaging (DTI) provides an in vivo measure of the microstructural development of WM and quantifies the integrity of WM in the brain as well as how this is affected by various factors. DTI studies of the brain help to better explain relationships between WM structure and language skills [12,13,14,15]. These studies have found several language-related DTI features. For example, asymmetries in the structure of the language network (the arcuate fasciculus) and diffusion parameters in the bilateral ventral WM pathways and corpus callosum have been associated with speech processing measures. In a longitudinal study conducted in 40 children with normal language development from infancy to kindergarten, Zuk et al. found that the white matter organization in infancy lays the foundation for long-term language development, particularly in the left arcuate tract and the left corticospinal tract, which are associated with phonological awareness [16]. Ekerdt et al. demonstrated neural plasticity in the white matter of typically developing preschool children as they learned new words. They also found that the white matter properties of the left middle temporal gyrus supported phonological processes [17]. Mitsuhashi et al. reported that the subcortical fiber networks of key brain regions, such as the right putamen and precentral gyrus, and the left superior temporal gyrus and supramarginal gyrus may contribute to the neuroplasticity of language [18]. Several other studies also have convincingly demonstrated the feasibility of using DTI to understand the neural mechanisms of language development [14,19].

Previous studies on the neural manifestations of language skills in typically developing children have generally been limited to children aged 8–15 years [20,21,22], but the relationship between WM microstructure and reading/language skills can manifest in early childhood. Due to difficulties related to data collection in preschool children, such as poor cooperation during MRI scanning and the presence of movement artifacts [23], there have been limited studies on the development of WM in preschool-aged children. The relationship between WM microstructure and specific pre-reading skills, such as phonological processing and speed naming, remains unclear in this age group. This age range is of particular interest due to the dynamic nature of behavioral, cognitive, and emotional regulation in young children aged between 2 and 7 [24].

These studies have confirmed the correlation between DTI characteristics and language development. However, no previous work has explained the neural mechanisms of language development in preschool children based on the number of fiber connections, and the underlying neural basis of successful phonological processing in children is still unknown. Therefore, this study explored the relationship between WM microstructure and phonological processing ability in the typical development of preschool-aged children.

The current study aimed to address these issues using the DTI data captured by the Calgary Preschool MRI Dataset, which comprises multimodal MRI data and relevant clinical data from preschoolers aged 2–7 years. We hypothesized that DTI-based changes to the brain microstructure affect phonological processing ability in preschool children. We investigated the correlation between DTI characteristics, age, and neuropsychological assessment scores and performed a mediation analysis to examine the inter-relationship between DTI-based dispersion tensor measures and the number of intersectional fiber connections in the brain as well as age and phonological processing test scores.

## 2. Materials and Methods

### 2.1. Participants

The present study is part of a wider project which uses multimodal MRI to investigate typical brain structural and functional development in early childhood. Reynolds et al. published the Calgary Preschool MRI dataset from this project [25], and we obtained 396 unprocessed DTI scans of 120 participants aged 2–7 years from the publicly available database. We excluded datasets that did not contain an associated T1 image and performed a visual inspection for the quality of the DTI data, ultimately preserving 208 datasets containing DTI and T1 data acquired in 73 children (38 male, 35 female), ranging in age from 1.95–6.97 years. All participants were born full-term (gestation period greater than 37 weeks) and had no diagnosed neurocognitive or developmental disorders. Their primary language was English and none of the children had received any professional reading tuition. Participants had no contraindications for undergoing MRI. The number of years of the mothers’ post-secondary education was used as a proxy for socioeconomic status (all mothers had completed high school) and ranged from 1 to 10 years. The children were asked to return for repeat testing approximately every six months, but not all children attended every appointment. Each time the children returned, they underwent an MRI scan and a neuropsychological assessment. The data included and analyzed in the present study include 22 children with one scan time point, 11 with two scans, 15 with three scans, 12 with four scans, 9 with five scans, 3 with six scans, and 1 with eight scans (Figure 1). Independent samples tests were used to examine sex differences in age, maternal education, and language skill (phonological processing and rapid naming) assessment scores. Table 1 summarizes the demographic characteristics and clinical scores of participants, including age, sex, maternal education, handedness, phonological processing raw score (PPRS), speed-naming completion time (SNCT), and speed-naming number correct (SNNC). The open database from which these data were accessed is available online: (article: https://doi.org/10.1016/j.dib.2020.105224; https://osf.io/axz5r/ (accessed on 6 June 2021)).

### 2.2. Neuropsychological Assessment

The neuropsychological assessment was performed on the same day as the MRI scan and took around 10 min to complete. Phonological processing and speed-naming subtests of the standardized developmental neuropsychological assessment 2nd Edition (NEPSY-II) [26] were used to assess children’s language skills. The phonological processing scaled score (PPRS) assesses a child’s ability to separate and process individual phonemes. When testing children aged 2 to 7, examiners point to three pictures and pronounce the corresponding words aloud. These words contain some identical phonemes and some different phonemes (e.g., (Duck, Dog, Door); (Fever, Believer)). When the examiner says a word, the child is asked to identify the corresponding picture. The speed-naming sub-test required rapid naming of colors and shapes. The children are asked to say the color and shape (e.g., circle and square) of the image as soon as possible, and the examiner scores the child based on the accuracy of the response (SNNC) and the time taken (SNCT).

The NEPSY-II provides a standardized test score for children aged 3–16 according to age. As some of the children recruited to this study were under 3 years of age, their original test scores were used in the subsequent analysis.

### 2.3. MRI Acquisition

Images were acquired using a GE 3T MR750w system with a 32-channel head coil (GE, Waukesha, WI) at the Alberta Children’s Hospital in Calgary, Canada. Before image acquisition, researchers conducted acclimatization procedures with participants to improve the quality of the data acquired in such young children. Specifically, children were given time to practice in a simulated scanner such that they could get used to the size and sound of the machine. Both the simulation scanners and MRI scanners had a space rocket appearance, and the children were told that they were astronauts on a mission. A detailed description is given in a previous article [27]. Participants were scanned either while they were awake watching a movie or sleeping.

A single-shot spin echo-planar imaging sequence was used to obtain a whole-brain diffusion-weighted image, with 1.6 × 1.6 × 2.2 mm^3^ resolution, reconstructed to 0.78 × 0.78 × 2.2 mm. Total brain coverage had a matrix size = 256 × 256 × 54, field of view = 200 mm × 200 mm, TR = 6750 ms, TE = 79 ms (in the first year of data collection, this was the TE as set to minimum), flip angle = 90°, anterior–posterior phase encoding, 30 diffusion encoding gradient directions at b = 750 s/mm^2^, and five interleaved images without diffusion encoding at b = 0 s/mm^2^. The scan duration was 4 min and 3 s.

### 2.4. DTI Data Processing

DTI data processing was conducted using DiffusionKit in bash, which provides a command-line interface to facilitate the batch processing of data [28]. DTI data preprocessing, the calculation of diffusion tensor indices, and brain network construction based on AAL90 were performed using the Processing Pipeline with the primary bash script available on the DiffusionKit website (accessed on 23 June 2021: https://diffusionkit.readthedocs.io/en/latest/tutorial_intro.html). DTI preprocessing comprised eddy current correction, skull dissection, registration, and normalization. Then fractional anisotropy (FA) and mean diffusivity (MD) of diffusion tensor indices were calculated, and the brain was divided into 90 regions of interest (ROIs) using the AAL90 template. For comparison with other studies, maps of FA and MD were registered to a standard pediatric template for children [29,30]. Finally, a brain network based on the number of fiber connections between brain regions was constructed using deterministic fiber tracking, and the brain structure network connection matrix (fiber number network, FN) was obtained. All processed images were visually checked to ensure high image quality with no artifacts or missing data.

### 2.5. Statistical Analysis

Statistical analysis was performed using SPM12, SPSS (IBM Statistics V25.0.0), and Python. The two variables in the correlation analysis were DTI measures based on 208 scans and 208 neuropsychological assessment scores obtained on the same day as the MRI scans. All correlational and mediation analyses controlled for the effects of within-subject longitudinal scan frequency, sex, handedness, and maternal education. We used the multiple regression feature of SPM12 to extract ROIs associated with age in the FA map and ROIs associated with PPRS in the MD map. The mean FA and MD values were calculated for each ROI. SPSS was used to calculate the correlation between the mean FA or mean MD value in each ROI with the age and PPRS as above. To further investigate phonological processing ability development, scripts written in Python were used to extract the 952 characteristics from the FN network (90 × 90), representing the number of fibrous connections in different brain regions. We then calculated the correlation between FN and age and neuropsychological assessment scores. Finally, we extracted 6 characteristics with significant correlations, where significance was considered at a corrected threshold of *p* < 0.05/952 = 5.25 × 10^−5^). Mediation analysis can be used to explore the underlying mechanisms of causation, helping researchers identify, formalize, and quantify possible mechanisms that associate a cause with an effect [31]. We performed mediation analyses following published methods [32,33], as detailed in our previous work [34], to evaluate the internal relationship between FN, mean FA, mean MD, age, and neuropsychological assessment scores. We assumed that age, PPRS, and DTI characteristics met the conditions for mediation analysis; that is, any two of them would be correlated. DTI characteristics included FA, MD, and the number of fibrous connections between brain regions. The mediating relationship we expected was that changes in the microstructural characteristics of DTI represented the underlying mechanism of phonological processing ability development with age. In other words, we needed to demonstrate that the mediation model “age→DTI characteristics→phonological processing ability development scores” holds true and that all other cases do not.

## 3. Results

### 3.1. Clinical and Behavioral Measures

The correlation analysis between age and the three neuropsychological assessments scores (PPRS, SNCT, and SNNC) found a significant positive correlation between age and PPRS (r = 0.731, *p* < 0.001, Figure 2A), SNCT (r = 0.465, *p* < 0.001, Figure 2B), and SNNC (r = 0.835, *p* < 0.001, Figure 2C). The SNCT of children aged between 2.5 and 3.5 years remained stable with small individual differences. SCNT decreased significantly at 3.5 years old and remained stable between 3.5 and 5 years with little difference in SNCT between individuals. At 5 years old, SNCT began to rise and showed large individual differences. Similarly, SNNC was stable between the ages of 2.5 and 5 years with small individual differences, while at 5 years of age, SNNC increased significantly and remained stable from 5 to 7 years.

There was a stronger correlation between age and PPRS than between either SNCT or SNNC and PPRS. PPRS increases gradually and continuously with age, while SNCT and SNNC remain relatively stable over a certain age range, followed by a jump increase. Therefore, in our follow-up analysis, PPRS was selected as the clinical scale most closely representing phonological processing ability.

### 3.2. Correlation between PPRS and DTI Measures

Clusters were found in the posterior limb of the internal capsule, the genu and body of the corpus callosum (CC), and the inferior fronto-occipital fasciculus that showed higher FA in older children (FA_Age+). Only a small area near the septum pellucidum body of the corpus callosum showed lower FA in older children (FA_Age−). A few clusters in the inferior fronto-occipital fasciculus and posterior limb of the internal capsule showed higher FA in individuals with higher PPRS (FA_PPRS+). There were no areas where higher FA was associated with lower PPRS.

MD showed almost the opposite pattern to that of FA. Only a small area close to the middle frontal gyrus showed higher MD in older children (MD_Age+). MD in almost all white matter areas, including the corpus callosum and the internal capsule, decreased with age (MD_Age−). Most regions including the internal capsule and corpus callosum showed lower MD in children with higher PPRS (MD_Age−). No region showed higher MD in older children.

In the FA map, a few clusters in the internal capsule and inferior fronto-occipital fasciculus were positively correlated with both age and PPRS (FA_Age+PPRS+, Figure 3A–C). In the MD map, there were clusters in the corpus collosum and internal capsule showing a negative correlation with age and a negative correlation with PPRS (MD_Age−PPRS−, Figure 3D–F). These regions exhibiting both positive correlations with age and PPRS in FA and negative correlations with age and PPRS in MD may represent phonological development in children over time. Thus, we performed a mediation analysis to examine the inter-relationship between the shared correlates (FA_Age+PPRS+orMD_Age−PPRS−), age, and PPRS. The results of this mediation analysis showed the model Age → FA_Age+PPRS+ → PPRS provided the best fit (Figure 4), suggesting that FA_Age+PPRS+ can lead an increase in PPRS with age. For completeness, we evaluated all 12 models (See Appendix A).

### 3.3. Associations between Language Performance, Age, and FN Characteristics

By further studying the FN matrix, we screened out six characteristics of the brain structure network (*p* < 5.25 × 10^−5^, see Section 2), which were vectors composed of the number of fiber connections in each pair of brain regions across all subjects. The six pairs of regions were: (1) right cuneus (CUN.R) and right calcarine fissure and surrounding cortex (CAL.R) (Figure 5A), (2) right calcarine fissure and surrounding cortex (CAL.R) and right lingual gyrus (LING.R) (Figure 5B), (3) left pallidum (PAL.L) and left olfactory cortex (OLF.L) (Figure 5C), (4) right superior occipital gyrus (SOG.R) and right inferior occipital gyrus (IOG.R) (Figure 5D), (5) right insula (INS.R) and right middle temporal gyrus (TPOmid.R) (Figure 5E), and (6) right inferior parietal lobule (IPL.R) and right supramarginal gyrus (SMG.R) (Figure 5F). Only one of these features is in the left hemisphere of the brain. Our hypothetical model was Age → FN → PPRS. The results of the mediation analysis of these six pairs of brain regions showed that only the number of fiber connections between IPL.R and SMG.R (IPL.R–SMG.R) led to the increase of PPRS with age, and the hypothesis was established (Figure 5F). The other five pairs of regions did not show significance (*p* ≥ 2.8 × 10^−^^7^), and the hypothesis was not rejected. For completeness, we evaluated all six models (See Appendix A). Figure 5 shows the relationship between the six characteristics and age and PPRS as well as the results of the partial mediation analysis.

## 4. Discussion

We investigated the inner relationship between structural WM changes in the brain and phonological processing ability development in an attempt to understand the underlying neural basis of behavioral performance in phonological and semantic processing in children aged 2–7 years. Correlation analysis of age and neuropsychological assessment scores showed that PPRS gradually increased with age, and phonological processing ability continued to develop and gradually improved from 2 to 7 years old. Studies have claimed that babies react differently to verbal and non-verbal sounds and familiar and unfamiliar languages in the first few days of life [3,35,36], which is consistent with our findings. This provides further evidence for the development of language being traceable back to the embryo, with humans learning and adapting to language at an early developmental stage. The speed and precision of the speed-naming subtest is a measure of the speed of language processing, reflecting connected speech, for example, the ability to produce a complete sentence [37]. In theory, an increased vocabulary could improve performance on a speed-naming test, but other factors, including cognition, memory, and attention, may also play a role. SNCT and SNNC remained stable before the age of 5 years, and individual differences were small. The stability of these scores means that there is little change in the rate of development of phonological processing skills before age 5. SNNC increased gradually between the ages of 5 to 7 years, while the SNCT increased markedly at 5 years old. These changes may be related to suddenly acquired or emergent behaviors, such as speaking a word for the first time after birth or speaking a complete sentence for the first time. These findings also highlight the importance of the language environment for phonological processing ability development after the age of 5 years.

These findings are consistent with the known developmental pattern of children [14,38,39,40]. By analyzing the associations between age and DTI measures, we replicated the finding that FA increases while MD decreases with age. These changes indicate a rapid increase in the rate of WM maturation in early childhood, reflecting either an increase in myelination or a change in axonal density or axon coherence [39]. The decrease in MD may partly reflect a decrease in the water content of the WM [41]. The results of our mediation analysis show that the development of FA is not the only factor contributing to phonological processing ability development in preschool children but that FA has a positive relationship with the development of phonological processing ability. Yeatman et al. [42] also confirmed that increased FA is indicative of better language ability.

The relationship between language or behavior and WM development is complex, with few longitudinal studies having been conducted. Most of the existing studies divide the brain’s structural fiber tracts into 10 separate WM tracts (the cingulum, corpus callosum body, corpus callosum genu, corpus callosum splenium, fornix, inferior fronto-occipital fasciculus, inferior longitudinal fasciculus, pyramidal, superior longitudinal fasciculus, and uncinate fasciculus) and study the relationships between the quantitative indices, such as FA and MD, of different tracts and language development. Consistent with our findings on linguistic structure networks, the present study suggests that linguistic networks may be more decentralized in children than in adults [43]. Therefore, the coarse separation of WM into 10 tracts is not precise and does not clarify the functional neuroanatomical features of language processing. Consequently, our study conducted a multivariate analysis based on the FN, which enabled us to clarify the relationship more exactly between WM structure and phonological processing ability development. To our knowledge, this is the first study using FN from DTI to study phonological processing ability development in young children (aged 2–7 years). We found that the increased number of fiber connections between the IPL and SMG in the right cerebral hemisphere was associated with an increased PPRS, which may be a key factor in language development in preschool-aged children. In conventional anatomical divisions, the supramarginal gyrus (SMG) is a part of the inferior parietal lobule (IPL). The classical neural language model considers the IPL as the main center of visual word recognition [44]. However, the IPL is heterogeneous in function and structure, which suggests that subregions of the IPL may have different contributions to reading. Its branch, the SMG, is thought to contribute preferentially to the phonological aspects of word processing [45]. In the AAL template, the IPL (excluding the parts of the SMG and angular gyrus/ANG that we classified as part of IPL directly) and SMG are two independent brain regions. Some functional neuroimaging studies support the importance of IPL in language function [46,47]. However, at present, there is no clear consensus on the functional role of IPL in language. IPL activation has been associated with the process of translating spelling into sound rather than the stored process of visual word spelling, suggesting that dysfunction in this area may be responsible for poor language skills in developmental dyslexics [48,49,50]. This is consistent with our findings.

Previous studies have shown that the brain regions associated with language are mainly located in the left hemisphere, but our results show that the most likely neural basis of a change related to phonological processing ability development is an increase in the number of fibers between the right hemisphere IPL and SMG. This suggests that the right hemisphere of the brain plays an important role in the typical language development of children, which reminds us that the neural mechanisms of language development in preschool children and adults are different. There is also the possibility that PPRS may only represent one aspect of language skill and therefore cannot be used in isolation to describe language function. Further, our findings, being based on objective measures of anatomical structure, differ from those of previous studies, which have been mostly based on fMRI and EEG measures of brain function.

However, the present study has some limitations. First of all, our imaging data and neuropsychological assessment scores were acquired longitudinally, with the same individual being invited back for multiple scans and assessments, but we treated each separate piece of data as an independent sample. Although we took participants’ ID as a covariable to remove the influence, it should still be noted that each separate piece of data is not the same as independent sample. Secondly, the present study was part of a wider study, whereby we only analyzed the inter-relationship between brain microstructural changes based on DTI and phonological processing, ignoring the rich information provided by other MRI contrast mechanisms. At the same time, phonological processing ability is only a very important part of language ability, and the neuropsychological subtests scores related to phonological processing adopted here cannot evaluate the language itself. To follow up on this, MRI data from scans using multiple contrast mechanisms and more neuropsychological scales related to language ability can be combined to better investigate the underlying neural mechanisms of language development in preschool children. Thirdly, mediation analysis is assuming a causal relationship, which needs to ensure that one thing occurs before the other. The mediating analysis of this study considered all possible models and proved that only the mediating model “age → brain microstructure change → development of phonological processing ability” was valid. However, the MRI was performed on the same day as the neuropsychological assessment, and the timing of the assessment does not support a true causal pathway. In subsequent studies, MRI data collection should be considered in advance to ensure the establishment of the hypothesized causal relationship. In addition, the correlations between regional DTI measures, age, and PPRS calculated were too scattered to be able to accurately describe these regions, and as such, only a few of the large correlating clusters were described in the paper. Another limitation of the study was that the children were scanned while sleeping or watching a movie in order to cooperate, but we counted both states as resting. Although the two states are different cognitive states, this study is based on the brain microstructure, and the changes of the brain structure in a short period of time can be ignored, which does not affect the structure-based findings. Finally, the sample tested in the present study included only typically developing healthy preschoolers and was limited in scope to exploring the relationship between brain white matter microstructure and phonological processing ability development. In future studies, participants with language development disorders can be included in pathological studies to explore the key imaging features associated with language development based on the neural mechanisms of typical language development.

## 5. Conclusions

Based on longitudinal DTI acquisition in preschool-aged children, we innovatively applied the brain fiber connection number matrix to study the relationship between the development of WM microstructure and phonological processing ability with age in early childhood. We found that the connection between the right inferior parietal lobule and supramarginal gyrus may have the potential to be a marker for phonological processing ability development. The findings of this study directly reflect the changes in brain microstructure and can help us to better understand the underlying neural mechanisms of language development in preschool children.

## Figures and Tables

**Figure 1 children-09-00782-f001:**
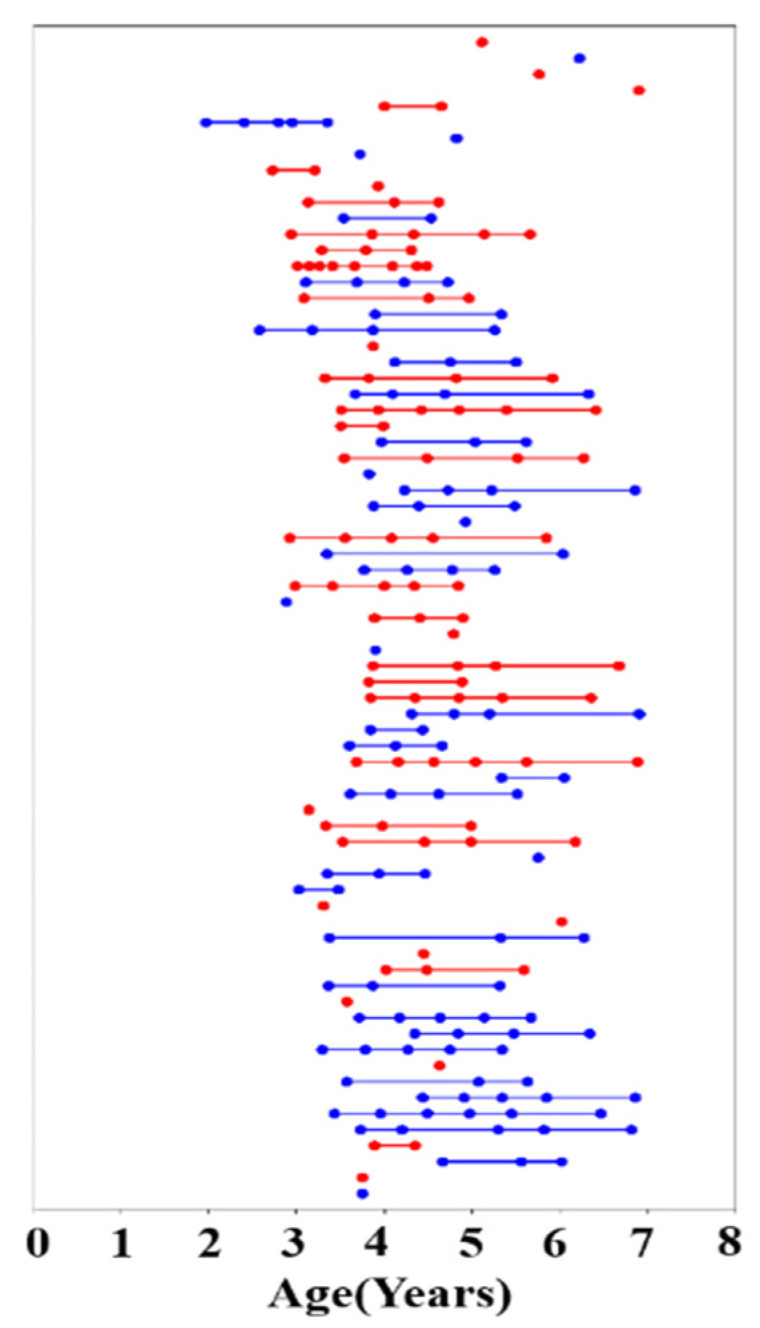
Age at time of scanning for 73 participants (red represents female, and blue represents male participants). Each scan is represented by a dot, and multiple scans on a given participant are connected by a straight line.

**Figure 2 children-09-00782-f002:**
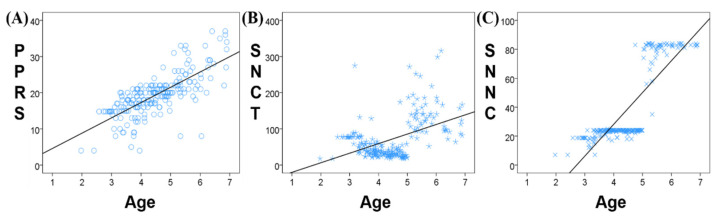
Correlation between age and neuropsychological assessment scores ((**A**) PPRS; (**B**) SNCT; (**C**) SNNC).

**Figure 3 children-09-00782-f003:**
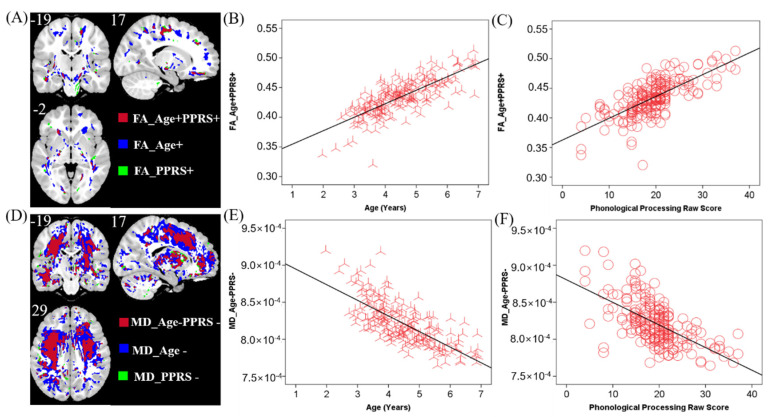
(**A**) Brain regions where FA positively correlated with age (blue, FA_Age+), PPRS (green, FA_PPRS+), and both (red, FA_Age+PPRS+). (**B**,**C**) Linear regression of FA_Age+PPRS+ vs. age (**B**) and PPRS (**C**). (**D**) MD negatively correlated with age (blue, MD_Age−), PPRS (green, MD_PPRS−), and both (red, MD_Age−PPRS−). (**E**,**F**) Linear regression of MD_Age−PPRS- vs. age (**E**) and PPRS (**F**). Clusters in A and D are overlaid on a T1 structural image. n = 208.

**Figure 4 children-09-00782-f004:**
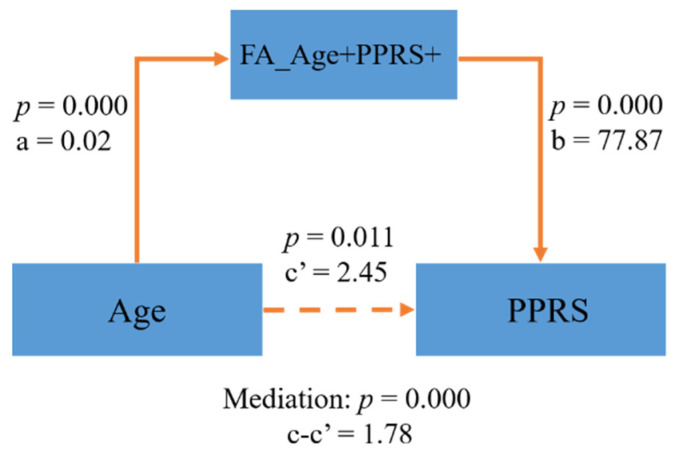
FA_Age+PPRS+ partly mediated the relationship of Age → PPRS.

**Figure 5 children-09-00782-f005:**
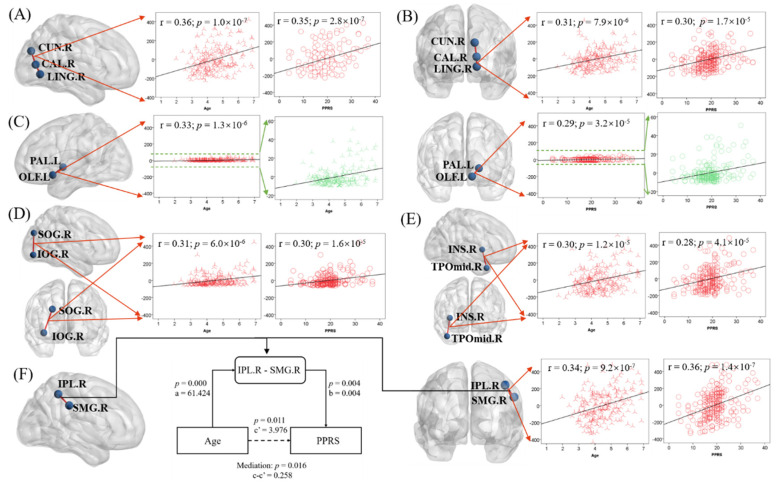
Linear regression of the fiber connection number in six pairs of regions ((**A**) CUN.R and CAL.R; (**B**) CAL.R and LING.R; (**C**) PAL.R and OLF.L; (**D**) SOG.R and IOG.R; (**E**) INS.R and TPOmid.R; (**F**) IPL.R and SMG.R) with age (**B**) and PPRS (**C**)). F also shows the results of the mediation analysis when the hypothetical model is Age → IPL. R-SMG.R → PPRS. Note that the scatter plots show the r and *p*, after age, sex, maternal education, and handedness had been accounted for.

**Table 1 children-09-00782-t001:** Demographics and Clinical Measures of Participants.

Characteristic	Male *n* = 38, m = 111	Female *n* = 35, m = 97	*t*	*p*
Age	4.56 ± 1.05	4.39 ± 0.97	−1.186	0.237
Maternal education	5.63 ± 2.35	5.62 ± 2.44	−0.32	0.975
Handedness (Left:Right:Both:N/A)	3:32:2:1	3:29:1:2	/	/
PPRS	19.47 ± 6.18	19.16 ± 5.51	−0.386	0.700
SNCT	79.18 ± 62.36	63.73 ± 50.99	−1.964	0.051
SNNC	41.74 ± 27.86	35.33 ± 25.00	−1.748	0.082

*n*, number of subjects; m, number of scans, rounded to two decimal places. N/A means that handedness had not yet been determined in these young children.

## Data Availability

The data that support the findings of this study are openly available at https://osf.io/axz5r/ accessed on 6 June 2021.

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
