# Peer review of "Associations between Brain Microstructure and Phonological Processing Ability in Preschool Children"

_children, 2022, doi:10.3390/children9060782_

Round 1

Reviewer 1 Report

The study aims at exploring the relationship between some aspects of language function and brain white matter microstructure based on DTI in healthy preschool subjects. Despite being the topic of interest theoretically, there are some major concerns that require attention and further explanation.

The study design is not very well explained. In particular, the reader understands along the paper that this study is part of a wider project, that is not clearly stated in the methods section. Despite mentioning a longitudinal design, a cross-sectional analysis is performed, but no clear detail of the number of subjects included is provided. Section 2.1 and 2.4 where the sample and data are described are inconsistent, presenting different numbers of subjects, different age ranges, with no description of the reason why the numbers are different. Also, the study design, clear hypothesis and procedures to support them are not well described.

No mention at all is about ethical approval of this specific study.

All the DTI related methodology is mostly based on self-citations. It seems a deterministic approach but this is not very well specified in the current paper.

Overall, the topic is of interest, however, the data and study design (for my understanding) do not allow the inferences the authors support, that this DTI approach can be predictive of dyslexia. There are no subjects with dyslexia or language disorders so I think this might eventually used as a future direction in pathological conditions. They use a single measure for language function, perform an association analysis), so the relationship between dyslexia and prediction is ambitious. It would be more realistic to support the relationship between language findings and dti results, by also comparing other studies by other groups and advanced DWI techinques in healthy controls or adults.

The lack of clarity on the study design and the data used (not clear in the current description how many subjects have multiple assessment and how many MRIs are from the same subject).

Another point which is not clear is how they obtained collaboration from younger infants 2-4, which is very hard to obtain. It would be of interest for the reader also for supporting the repeatability of the study.

Specific comments.

Abstract. To be revised, better specify where dti metrics correlated with clinical measures. No prediction can be done as an association analysis is performed (but not clear, as explained in the general comments), the study do not provide evidences for early diagnosis or intervention in dyslexia, so this type of inference should be removed or substantially toned down (if properly described in the discussion), but again, not subjects with language disorder or dyslexia are included here.

Introduction: should be focused on language development, the relevance of the measures used for language evaluation (limited) and on DTI and findings in relationship with language function. I suggest to include a scenario on DTI techniques and analysis to allow the understanding of the role and accuracy of the one they use. No pathology should be mentioned here as the study is in typical development.

Materials and methods

Clear specify the study design, that the study is part of a wider one, include ethical statement. Clarify the number of subjects, of clinical and MRI evaluation for the same subject.

Discussion

Should be amended according to the general comments and changes in previous sections.

Reviewer 2 Report

In the manuscript titled "Associations between Brain Microstructure and Language Ability in Preschool Children" Ying Zhou and colleagues, they have reported that correlation analysis showed that across the whole age group, with increasing age, Phonological Processing Raw Score increased, fractional anisotropy increased, and mean diffusivity decreased. The results of the mediation analysis suggest that increased fractional anisotropy may be the basis of language skill development during this period, and the increased number of fiber connections between the right inferior parietal lobule and right supramarginal gyrus may predict the development of language ability. This connection may be a promising marker for language development. This study provides objective evidence for diagnosis and early intervention in dyslexia. I have a few comments regarding the present manuscript.

-The manuscript is an interesting and well-design study, however more detailed information about how the samples were processed is necessary, sections 2.3 and 2.4

-Both variables SNCT and SNNC are almost different, the authors have used a corrected statistical model to fix that?

-More detailed information about what means mediation analysis is required

-The results section and discussion look well. Thanks to the authors for adding limitations.

Round 2

Reviewer 2 Report

Thank you to the authors for taking into account my previous comments.

Only a minor comment about the reference style, please check the reference guidelines.
